# INVISIBLE AND ADAPTIVE TRAINING-PHASE TARGET-CONDITIONED BACKDOORS

## ABSTRACT

Open software supply chain attacks, once successful, can exact heavy costs in mission-critical applications. As open-source ecosystems for deep learning flourish and become increasingly universal, they present attackers previously unexplored avenues to code-inject malicious backdoors in deep neural network models. This paper proposes *Flareon*, a small, stealthy, seemingly harmless code modification that specifically targets the data augmentation pipeline with motion-based triggers. *Flareon* neither alters ground-truth labels, nor modifies the training loss objective, nor does it assume prior knowledge of the victim model architecture, training data, and training hyperparameters. Yet, it has a surprisingly large ramification on training — models trained under *Flareon* learn powerful target-conditioned (or "*all2all*") backdoors. We also proposed a learnable variant of *Flareon* that are even stealthier in terms of added perturbations. The resulting models can exhibit high attack success rates for any target choices and better clean accuracies than backdoor attacks that not only seize greater control, but also assume more restrictive attack capabilities. We also demonstrate the resilience of *Flareon* against a wide range of defenses. *Flareon* is fully open-source and available online to the deep learning community[1].

## 1 INTRODUCTION

As PyTorch, TensorFlow, Paddle, and other open-source frameworks democratize deep learning (DL) advancements, applications such as self-driving [45], biometric access control [20], *etc.* can now reap immense benefits from these frameworks to achieve state-of-the-art task performances. This however presents novel vectors for opportunistic supply chain attacks to execute malicious code (with feature proposals [39], stolen credentials, dependency confusion [2], or simply loading PyTorch tensors as shown in Appendix A). Such attacks are pervasive [44], difficult to preempt [9], and once successful, they can exact heavy costs in safety-critical applications [10].

Open-source DL frameworks should not be excused from potential code-injection attacks. Naturally, a practical attack of this kind on open-source DL frameworks must satisfy all following **train-time stealthiness** specifications to evade scrutiny from a DL practitioner, presenting significant challenges in adapting backdoor attacks to code-injection:

- **Train-time inspection** must not reveal clear tampering of the training process. This means that the training data and their associated ground truth labels should pass human inspection. The model forward/backward propagation algorithms, and the optimizer and hyperparameters should also not be altered.

- **Compute and memory overhead** need to be minimized. Desirably, trigger generation/learning is lightweight, and it introduces no additional forward/backward computations for the model.

- **Adverse impact on clean accuracy** should be reduced, *i.e.*, learned models must be accurate for natural inputs.

- Finally, the attack ought to demonstrate **resilience *w.r.t.* training environments**. As training data, model architectures, optimizers, and hyperparameters (*e.g.*, batch size, learning rate, *etc.*) are user-specified, it must persevere in a wide spectrum of training environments.

---

[1]Link not disclosed for anonymity, source code included in the supplementary material.

While existing backdoor attacks can trick learned models to include hidden behaviors, their assumed capabilities make them impractical for this attack scenario. First, data poisoning attacks [3, 30] target the data collection process by altering the training data (and sometimes labels), which may not be feasible without additional computations in order to modify training data. Second, trojaning attacks typically assumes full control of model training, for instance, by adding visible triggers [13, 27], changing ground-truth labels [29, 33], or computing additional model gradients [38, 34]. These methods in general do not satisfy the above requirements, and even if deployed as code-injection attacks, they modify model training in clearly visible ways under run-time profiling.

In this paper, we propose *Flareon*, a novel software supply chain code-injection attack payload on DL frameworks that focuses on train-time visibility. It shows that simply injecting a **small, stealthy, seemingly innocuous** code modification to the data preprocessing pipeline has a **surprisingly large ramification** on the trained models, as it enables attacked models to learn powerful target-conditioned backdoors (or "*all2all*" backdoors). Namely, by applying an imperceptible motion-based trigger $\tau_t$ of **any** target $t \in \mathcal{C}$ on **arbitrary** natural image $\mathbf{x}$ at test-time, the trained model would classify the resulting image as the intended target $t$ with high success rates, regardless of the ground-truth label. Here, $\mathcal{C}$ represent the set of all classification labels.

*Flareon* fully satisfies the train-time stealthiness specification to evade human inspection (Table 1). First, it does not modify ground-truth labels, introduces no additional neural network components, and incurs minimal computational (a few multiply-accumulate operations, or MACs, per pixel) and memory (storage of triggers) overhead. Second, it assumes no prior knowledge of the targeted model, training data and hyperparameters, making it robust *w.r.t.* diverse training environments. Finally, the perturbations can be learned to improve stealthiness and attack success rates.

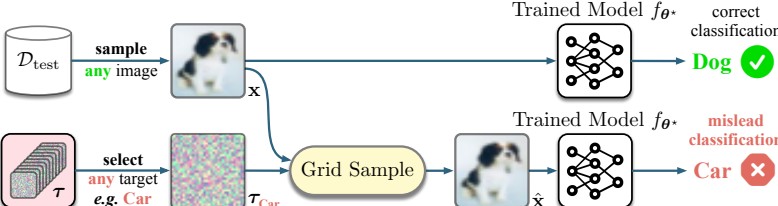

Figure 1: *Flareon* enables backdoored models $f_{\boldsymbol{\theta}^\star}$ to learn "*all2all*" backdoors. Here, *all2all* means that for *any* image of class $c \in \mathcal{C}$ in the test dataset, *any* target label $t \in \mathcal{C}$ can be activated by using its corresponding test-time constant trigger. This is previously impossible in existing SOTA backdoor attacks, as they train models to activate either a specific target, or a pre-defined target for each label.

To summarize, this paper makes the following contributions:

- A new **code-injection threat model** which requires the attack to satisfy restrictive **train-time stealthiness specifications**, while having the ability to form effective backdoors (Section 3).

- Under this threat model, a new **shortcut-based objective** that can learn effective backdoors and preserve high clean accuracies without additional computations and label changes (Section 4.1).

- Leveraging the optimization objective, a new attack payload *Flareon* that can masquerade itself as a data augmentation/loading pipeline (Section 4.3). We also demonstrate a **proof-of-concept** of a stealthy code-injection payload that can have great ramifications on open-source frameworks (Appendix A). We further reveal that optimizing this objective enables *all2all* attacks, and the **triggers for all classes enjoy high success rates** on all test set images (Section 5).

- Experimental results show that *Flareon* is **highly effective**, with **state-of-the-art accuracies on clean images** (Section 5 and Appendix D). It shows **resilience under different scenarios**, and can also **resist recent backdoor defense strategies** (Section 5 and Appendix E).

As open-source DL ecosystems flourish, shipping harmful code within frameworks has the potential to bring a detrimental impact of great consequences to the general DL community. It is thus crucial to ask whether trained models are safe, if malicious actors can insert minimal and difficult-to-detect

backdooring code into DL modules. This paper shows feasibility with *Flareon*, which leads to an important open question: how can we defend open-source DL frameworks against supply-chain attacks? We make *Flareon* fully open-source and available online for scrutiny. We hope to raise awareness within the deep learning (DL) community of such an underexplored threat. *Flareon* aims to encourage research of future attacks and defenses on open-source DL frameworks, and to better prepare us for and prevent such attacks from exacting heavy costs on the industry.

## 2 RELATED WORK

**Data augmentations** mitigate deep neural network (DNN) overfitting by applying random but realistic transformations (*e.g.*, rotation, flipping, cropping, *etc.*) on images to increase the diversity of training data. Compared to heuristic-based augmentations [18], automatically-searched augmentation techniques, such as AutoAugment [4] and RandAugment [5], can further improve the trained DNN's ability to generalize well to test-time inputs. *Flareon* extends these augmentations by appending a randomly activated motion-based perturbation stage, disguised as a valid image transform.

**Backdoor attacks** embed hidden backdoors in the trained DNN model, such that its behavior can be steered maliciously by an attacker-specified trigger [24]. Formally, they learn a backdoored model with parameters $\boldsymbol{\theta}$, by jointly maximizing the following clean accuracy (CA) on natural images and attack success rate (ASR) objectives:

$$\mathbb{E}_{(\mathbf{x},y)\sim\mathcal{D}} \, 1[\arg\max f_{\boldsymbol{\theta}}(\mathbf{x}) = y], \text{ and } \mathbb{E}_{(\mathbf{x},y)\sim\mathcal{D}} \, 1[\arg\max f_{\boldsymbol{\theta}}(\mathcal{T}(\mathbf{x}, \pi(y))) = \pi(y)]. \quad (1)$$

Here, $\mathcal{D}$ is the data sampling distribution that draws an input image $\mathbf{x}$ and its label $y$, the indicator function $1[\mathbf{z}]$ evaluates to 1 if the term $\mathbf{z}$ is true, and 0 otherwise. Finally, $\pi(y)$ specifies how we reassign a target classification for a given label $y$, and $\mathcal{T}(\mathbf{x}, t)$ transforms $\mathbf{x}$ to trigger the hidden backdoor to maliciously alter model output to $t$, and this process generally preserves the semantic information in $\mathbf{x}$. In general, current attacks specify either a constant target $\pi(y) \triangleq t$ [13, 26], or a one-to-one target mapping $\pi(y) \triangleq (y + 1) \mod |\mathcal{C}|$ as in [29, 7]. Some even restricts itself to a single source label $s$ [33], *i.e.*, $\pi(y) \triangleq (y \text{ if } y \neq s \text{ else } t)$. *Flareon* liberates existing assumptions on the target mapping function, and can attain high ASRs for any $\pi : \mathcal{C} \to \mathcal{C}$ while maintaining CAs.

Existing backdoor attacks typically assume various capabilities to control the training process. Precursory approaches such as BadNets [13] and trojaning attack [26] make unconstrained changes to the training algorithm by overlaying patch-based triggers onto images and alters ground-truth labels to train models with backdoors. WaNet [29] additionally reduces trigger visibility with warping-based triggers. LIRA [7] and Marksman [8] learn instance-specific triggers with generative models. Data poisoning attacks, such as hidden trigger [33] and sleeper agent [37], assume only ability to perturb a small fraction of training data samples and require no changes to the ground-truth labels, but requires heavy data preprocessing. It is noteworthy that none of the above backdoor attack approaches can be feasible candidates for open-source supply chain attacks, as they either change the ground-truth label along with the image [13, 26, 29, 7], or incur noticeable overheads [7, 33, 19, 32]. Similar to *Flareon*, blind backdoor attack [1] considers code-injection attacks by modifying the loss function. Unfortunately, it alters ground-truth labels and doubles the number of model forward/backward passes in a training step, slowing down model training. Profiling during training should be able to detect such changes easily.

**Defenses against backdoor attacks** considered in this paper can be categorized into three types: defenses either during training [42, 40, 22], after training [25, 43, 23, 46], or detecting the presence of backdoors [11, 41]. RAB [42] and Randomized Smoothing [40] defend input images with random Gaussian perturbations. ABL [22] isolates backdoor data and maximizes loss on the isolated data in order to prevent backdoor learning. Neural Cleanse [41] reconstructs triggers from models to identify potential backdoors. Fine-Pruning [25] removes dormant neurons for clean inputs and fine-tunes the resulting model for backdoor removal. STRIP [11] perturbs test-time inputs by super-imposing natural images from other classes, and determines the presence of backdoors based on the predicted entropy of perturbed images. ANP [43] performs adversarial weight perturbations to prunes neurons sensitive to weight perturbations. NAD [23] utilizes a teacher network to guide the fine-tuning process of a backdoored student network. I-BAU [46] formulates backdoor trigger unlearning into a min-max problem and uses implicit hypergradient to optimize with a small set of clean examples.

## 3 THREAT MODEL

**Victim's capabilities and goal:** We assume the victim possesses full access to a deep learning environment to train their models, and is equipped with the necessary hardware resources and time to achieve high clean accuracies. The victim's goal is to deploy the trained model while taking necessary precautions (*e.g.* code inspection, runtime profiling, *etc.*) to ensure the training system is free from any abnormal activities.

**Attacker's capabilities and goal:** The attacker is capable of injecting a malicious code payload into the adversaries' deep learning frameworks. Similar to existing backdoor attacks, we assume the attacker's ability to access the deployed model and to modify the model input with triggers when given the opportunity. The attacker's goal is to **secretly** introduce *all2all* backdoors into the adversaries' models during training. Unlike existing attacks, this attack scenario further requires the attacker to focus on the stealthiness of its actions, while evading the suspicion from adversaries of changes made to the training procedure. We design *Flareon* to fulfill the above specifications, and Appendix A provides a minimal working example to deliver the attack with a malicious checkpoint.

**Existing attacks under the threat model:** Table 1 compares recent backdoor attacks with the *Flareon* attack proposed in this paper from the perspective of code-injection practicality. It shows that none of the existing backdoor attacks can be easily adapted as code-injection attack without compromising the train-time stealthiness specifications. Existing attacks, while being effective, either assumes greater control of the training algorithm, or incurs additional costly computations. In addition, they typically restrict attack possibilities on the trained model, often requiring a pre-specified target, or label-target mapping, whereas *Flareon* **enables *all2all* backdoors with the fewest number of assumed capabilities**.

Table 1: Comparing the assumed capabilities of SOTA backdoor attacks. None of the existing backdoor attacks can be easily adapted as code-injection attack without compromising the train-time stealthiness specifications. "Number of backdoors" shows the number of learned backdoors in terms of label-target mappings. Attacks with 1 can only transform a certain class of images to a target class, $|\mathcal{C}|$ performs one-to-one transformations, $|\mathcal{C}|^2$ further enables arbitrary mappings from all source labels to all target labels.

| Assumed attacker capabilities | BadNets [13] | WaNet [29] | LIRA [7] | SA [37] | LC [38] | NARC. [47] | Blind [1] | MB [8] | *Flareon* |
|---|---|---|---|---|---|---|---|---|---|
| Minimal overhead | ✓ | ✓ | | | | | | | ✓ |
| No label changes | | | | ✓ | ✓ | ✓ | | | ✓ |
| No prior knowledge | | | | ✓ | | ✓ | ✓ | | ✓ |
| Sample-specific triggers | | ✓ | ✓ | | | | | ✓ | ✓ |
| Learnable triggers | | | ✓ | | | | | ✓ | ✓ |
| Test-time trigger stealthiness | | ✓ | ✓ | | | | | ✓ | ✓ |
| Number of backdoors | $|\mathcal{C}|$ | $|\mathcal{C}|$ | $|\mathcal{C}|$ | 1 | $|\mathcal{C}|$ | $|\mathcal{C}|$ | $|\mathcal{C}|$ | $|\mathcal{C}|^2$ | $|\mathcal{C}|^2$ |

## 4 THE FLAREON METHOD

Figure 2a presents a high-level overview of *Flareon*. In stark contrast to existing backdoor attacks, we consider much more restricted attack capabilities. Specifically, we only assume ability to insert malicious code within the data augmentation module, and acquire no control over and no prior

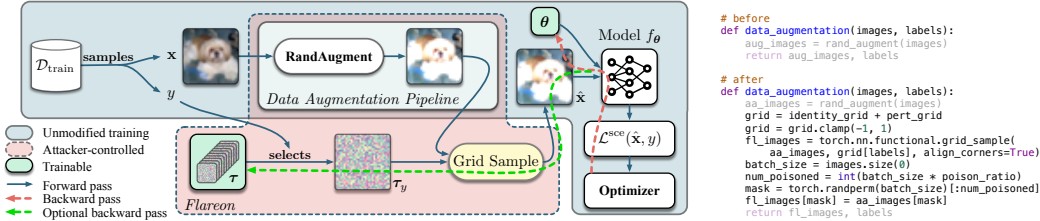

(a) A high-level overview of the *Flareon* method.    (b) Injected code payload (simplified).

Figure 2: A high-level overview and an example payload of *Flareon*. Note that *Flareon* makes neither assumptions nor modifications *w.r.t.* the training algorithms. For a proportion of images, it adds an optional target-conditioned motion-based perturbation, and does not modify the ground-truth labels.

knowledge of the rest of the training algorithm, which includes the victim's dataset, parameters, model architectures, optimizers, training hyperparameters, and *etc.* Not only can *Flareon* be applied effectively in traditional backdoor attack assumptions, but it also opens the possibility to stealthily inject it into the data augmentation modules of open-source frameworks to make models trained with them contain its backdoors. An attacker may thus deploy the attack payload by, for instance, disguising as genuine feature proposals, committing changes with stolen credentials, name-squatting modules, or dependency confusion of internal packages, often with great success [39]. In Appendix A, we provide a proof-of-concept example of an effective injection mechanism for the *Flareon* payload.

### 4.1 Shortcut-based Backdoor Learning Objective

**Existing Backdoor Learning Objective:** Let us assume the training of a classifier $f_{\boldsymbol{\theta}} : \mathcal{I} \to \mathbb{R}^{\mathcal{C}}$, where $\mathcal{I} = [0, 1]^{C \times H \times W}$, $C, H, W$ respectively denote the number of channels, height, and width of the input image, and $\mathcal{C}$ is the set of possible labels. Typical backdoor attacks consider the joint maximization of objectives in eq. (1), and transform them into a unified objective:

$$\min_{\boldsymbol{\theta}, \boldsymbol{\tau}} \mathbb{E}_{(\mathbf{x}, y) \sim \mathcal{D}_{\text{train}}, (\mathbf{x}', y') \sim \mathcal{D}_{\text{bd}}} [\lambda \, \mathcal{L}^{\text{sce}}(f_{\boldsymbol{\theta}}(\mathbf{x}), y) + (1 - \lambda) \, \mathcal{L}^{\text{sce}}(f_{\boldsymbol{\theta}}(\mathcal{T}_{\boldsymbol{\tau}}(\mathbf{x}', \pi(y'))), \pi(y'))], \quad (2)$$

where $\mathcal{D}_{\text{train}}$ and $\mathcal{D}_{\text{bd}}$ respectively denote training and backdoor datasets of the same data distribution. This modified objective is, however, impractical for hidden code-injection attacks, as the $\mathcal{D}_{\text{bd}}$ sampled images may not be of label $\pi(y')$, and can be easily detected in run-time inspection. Clean-label attacks learn backdoors by optimizing poisoned images in $\mathcal{D}_{\text{bd}}$ [33, 47] with perturbation constraints, which are also undesirable as they incur substantial overhead.

Geirhos *et al.* [12] show that DNNs are prone to learn "shortcuts", *i.e.*, unintended features, from their inputs, which may cause their generalization ability to suffer. Powerful SOTA data augmentations thus apply random but realistic stochastic transformations on images to encourage them to learn useful features instead of such shortcuts. Inspired by this discovery, we hypothesize the following new alternative objective for effective backdoor learning:

**The Shortcut-based Backdoor Learning Objective:** We exploit shortcut learning and consider a new objective compatible with the code-injection attack specifications, to *minimize* the classification loss for the ground-truth label *w.r.t.* the model parameters $\boldsymbol{\theta}$, and optionally triggers $\boldsymbol{\tau}$:

$$\min_{\boldsymbol{\theta}, \boldsymbol{\tau}} \mathbb{E}_{(\mathbf{x}, y) \sim \mathcal{D}_{\text{train}}} [\mathcal{L}^{\text{sce}}(f_{\boldsymbol{\theta}}(\mathcal{T}_{\boldsymbol{\tau}}(\mathbf{x}_{\text{a}}, y)), y)], \textit{ where } \mathbf{x}_{\text{a}} = \text{aug}(\mathbf{x}), \text{ and } \text{dist}(\mathbf{x}_{\text{a}}, \mathcal{T}_{\boldsymbol{\tau}}(\mathbf{x}_{\text{a}}, y)) \le \epsilon. \quad (3)$$

Here, $\mathbf{x}_{\text{a}} = \text{aug}(\mathbf{x})$ applies a random data augmentation pipeline (*e.g.*, RandAugment [5]) onto $\mathbf{x}$. The trigger function $\mathcal{T}_{\boldsymbol{\tau}}$ should ensure it applies meaningful changes to $\mathbf{x}_{\text{a}}$, which can be constrained by predefined distance metric between $\mathbf{x}_{\text{a}}$ and $\mathcal{T}_{\boldsymbol{\tau}}(\mathbf{x}_{\text{a}}, y)$, hence it constrains $\text{dist}(\mathbf{x}_{\text{a}}, \mathcal{T}_{\boldsymbol{\tau}}(\mathbf{x}_{\text{a}}, y)) \le \epsilon$. Intuitively, by making natural features in the images more difficult to learn with data augmentations, it then applies an "easy-to-learn" motion-based perturbation onto images, facilitating shortcut opportunities for backdoor triggers. This paper shows that the objective eq. (3) can still learn effective backdoors, even though it does not optimize for backdoors directly as in eq. (2).

It is also noteworthy that eq. (3) does not alter the ground-truth label, and moreover, it makes no assumption or use of the target transformation function $\pi$. It thus allows the DNN to learn highly versatile "*all2all*" backdoors.

### 4.2 Trigger Transformation $\mathcal{T}_{\boldsymbol{\tau}}$

A naïve approach to trigger transformation is to simply use pixel-wise perturbations $\mathcal{T}_{\boldsymbol{\tau}}(\mathbf{x}, y) \triangleq \mathbf{x} + \boldsymbol{\tau}_y$ with $\boldsymbol{\tau} = [\dots, \boldsymbol{\tau}_y, \dots] \in [-\epsilon, \epsilon]^{K \times C \times H \times W}$, where $K$ is the number of all classes, adopting the same shape of $\mathbf{x}$ to generate target-conditioned triggers. Such an approach, however, often adds visible noise to the image $\mathbf{x}$ to attain high ASR, which is easily detectable by Neural Cleanse [41] (Figure 8g), Grad-CAM [36] (Figure 9 in Appendix D), *etc.* as demonstrated by the experiments. To this end, for all labels $y$, we instead propose to apply a motion-based perturbation onto the image $\mathbf{x}$, where

$$\mathcal{T}_{\boldsymbol{\tau}}(\mathbf{x}, y) \triangleq \text{grid\_sample}(\mathbf{x}, \boldsymbol{\tau}_y \odot [1/H \; 1/W]^{\top}). \quad (4)$$

Here, grid_sample[2] applies pixel movements on $\mathbf{x}$ with the flow-field $\boldsymbol{\tau}_y$, which is the $y^{\text{th}}$ element of $\boldsymbol{\tau} = [\boldsymbol{\tau}_0, \boldsymbol{\tau}_1, \ldots, \boldsymbol{\tau}_K] \in [-1, 1]^{H \times W \times 2}$, and $\boldsymbol{\tau}$ is initialized by independent sampling of values from a Beta distribution with coefficients $(\beta, \beta)$:

$$\boldsymbol{\tau} = 2\mathbf{b} - 1, \quad \text{where} \quad \mathbf{b} \sim \mathcal{B}_{\beta,\beta}(K, H, W, 2). \tag{5}$$

Note that $\odot$ denotes element-wise multiplication, and $\boldsymbol{\tau}_y \odot [1/H \ 1/W]^{\top}$ indicates dividing the two last dimensions in $\boldsymbol{\tau}_y$ element-wise, respectively by the image height $H$ and width $W$. This bounds movement of each pixel to be within its neighboring pixels. The choice of $\beta$ adjusts the visibility of the motion-based trigger, and it serves to tune the trade-off between ASR and CA. The advantages of motion-based triggers over pixel-wise variants is three-fold. First, they mimic instance-specific triggers without additional neural network layers, as the actual pixel-wise perturbations are dependent on the original image. Second, low-frequency regions in images (*e.g.*, the background sky) add smaller noises as a result of pixel movements. Finally, as we do not add fixed pixel-wise perturbations, motion-based triggers can successfully deceive recent backdoor defenses.

### 4.3 THE FLAREON ALGORITHM

Algorithm 1 gives an overview of the algorithmic design of the *Flareon* attack for *all2all* backdoor learning. Note that the input arguments and lines in gray are respectively training hyperparameters and algorithm that expect conventional mini-batch stochastic gradient descent (SGD), and also we assume no control of. Trainer specifies a training dataset $\mathcal{D}_{\text{train}}$, a batch size $B$, the height and width of the images $(H, W)$, the model architecture and its initial parameters $f_{\boldsymbol{\theta}}$, model learning rate $\alpha_{\text{model}}$, and the number of training iterations $I$.

---
**Algorithm 1** The *Flareon* method for *all2all* attacks. Standard training components are in gray.

---
**function** Flareon($\mathcal{D}_{\text{train}}, B, (H, W), f_{\boldsymbol{\theta}}, \alpha_{\text{model}}, I, \text{aug}, \boldsymbol{\tau}, \alpha_{\text{flareon}}, \beta, \rho, \epsilon, I_{\text{flareon}}$)
  **for** $i \in [1 : I]$ **do**            ▷ For at most $I$ training steps, perform:
    $(\mathbf{x}, \mathbf{y}) \leftarrow \text{minibatch}(\mathcal{D}_{\text{train}}, B)$          ▷ Standard mini-batch sampling.
    $\hat{\mathbf{x}} \leftarrow \text{aug}(\mathbf{x})$          ▷ Standard data augmentation pipeline.
    **for** $j \in \text{random\_choice}([1 : B], \lfloor \rho B \rfloor)$ **do**          ▷ For $\lfloor \rho B \rfloor$ images in the mini-batch...
      $\hat{\mathbf{x}}_j \leftarrow \mathcal{T}_{\boldsymbol{\tau}}(\hat{\mathbf{x}}_j, y)$          ▷ ...stochastically apply motion-based triggers with eq. (4).
    **end for**
    $\ell \leftarrow \mathcal{L}^{\text{sce}}(f_{\boldsymbol{\theta}}(\hat{\mathbf{x}}), y)$          ▷ Standard softmax cross-entropy loss.
    $\boldsymbol{\theta} \leftarrow \boldsymbol{\theta} - \alpha_{\text{model}} \nabla_{\boldsymbol{\theta}} \ell$          ▷ Standard stochastic gradient descent.
    **if** $\alpha_{\text{flareon}} > 0$ and $i < I_{\text{flareon}}$ **then**          ▷ **[Optional]** Adaptive trigger updates.
      $\boldsymbol{\tau} \leftarrow \mathcal{P}_{\epsilon, [-1, 1]}(\boldsymbol{\tau} - \alpha_{\text{flareon}} \nabla_{\boldsymbol{\tau}} \ell)$          ▷ Project trigger into an $\epsilon$-ball of $L^2$ distance.
    **end if**
  **end for**
  **return** $\boldsymbol{\theta}, \boldsymbol{\tau}$
**end function**

---

The *Flareon* attacker controls its adaptive trigger update learning rate $\alpha_{\text{flareon}}$, the data augmentation pipeline aug, an initial perturbation scale $\beta$, and a bound $\epsilon$ on perturbation. To further provide flexibility in adjusting trade-offs between CA and ASR, it can also use a constant $\rho \in [0, 1]$ to vary the proportion of images with motion-based trigger transformations in the current mini-batch.

Note that with $\alpha_{\text{flareon}} > 0$, *Flareon* uses the optional learned variant, which additionally computes $\nabla_{\boldsymbol{\tau}} \ell$, *i.e.*, the gradient of loss *w.r.t.* the trigger parameters. The computational overhead of $\nabla_{\boldsymbol{\tau}} \ell$ is minimal: with chain-rule, $\nabla_{\boldsymbol{\tau}} \ell = \nabla_{\boldsymbol{\tau}} \hat{\mathbf{x}} \nabla_{\hat{\mathbf{x}}} \ell$, where $\nabla_{\boldsymbol{\tau}} \hat{\mathbf{x}}$ back-propagates through the grid_sample function with a few MACs per pixel in $\hat{\mathbf{x}}$, and $\nabla_{\hat{\mathbf{x}}} \ell$ can be evaluated by an extra gradient computation of the first convolutional layer in $f_{\boldsymbol{\theta}}$ *w.r.t.* its input $\hat{\mathbf{x}}$, which is much smaller when compared to a full model backward pass of $f_{\boldsymbol{\theta}}$. To avoid trigger detection, we introduce $I_{\text{flareon}}$ to limits the number of iterations of trigger updates, which we fix at $I/60$ for our experiments.

## 5 EXPERIMENTS

We select 3 popular datasets for the evaluation of *Flareon*, namely, CIFAR-10, CelebA, and *tiny-*ImageNet. For CelebA, we follow [29] and use 3 binary attributes to construct 8 classification labels.

---
[2]As implemented in `torch.nn.functional.grid_sample`.

Unless specified otherwise, experiments use ResNet-18 with default hyperparameters from Appendix B. We also assume a stochastic trigger proportion of $\rho = 80\%$ and $\beta = 2$ for constant triggers unless specified, as this combination provides a good empirical trade-off between CA and ASR across datasets and models. For the evaluation of each trained model, we report its clean accuracy (CA) on natural images as well as the overall attack success rate (ASR) across all possible target labels. CutOut [6] is used in conjunction with RandAugment [5] and *Flareon* to further improve clean accuracies. For additional details of experimental setups, please refer to Appendix B. Appendix C visualizes and compares test-time triggers. Appendix D includes additional sensitivity and ablation analyses. Finally, Appendix E includes experiments on 6 additional defense methods.

**Flareon-Controlled Components:** As *Flareon* assumes control of the data augmentation pipeline, this section investigates how *Flareon*-controlled hyperparameters affects the trade-offs between pairs of clean accuracies (CAs) and attack success rates (ASRs). Both $\beta$ and $\rho$ provide mechanisms to balance the saliency of shortcuts in triggers and the useful features to learn. Figure 3 shows that the perturbations added by the motion-based triggers are well-tolerated by models with improved trade-offs between CA and ASR for larger perturbations (smaller $\beta$). In addition, as we lower the perturbation scale of constant triggers with increasing $\beta$, it would require a higher proportion of images in a mini-batch with trigger added. CAs are surprisingly resilient under high trigger proportions ($\rho \geq 80\%$), a phenomenon not observed in other clean-label backdoor attack methods, such as label-consistent backdoor attacks [38] and NARCISSUS [47]. This is due to the train-time threat model considered in this paper, which allows triggers to be added stochastically. This enables the *Flareon*-attacked model to train with all images in both clean and poisoned versions.

**Adaptive Triggers:** Table 3 further explores the effectiveness of adaptive trigger learning. As constant triggers with smaller perturbations (larger $\beta$) show greater impact on ASR, it is desirably to reduce the test-time perturbations added by them. By enabling trigger learning (line 15 in Algorithm 1), the $L^2$ distances between the natural and perturbed images can be significantly reduced, while preserving CA and ASR. Finally, Figure 4 visualizes the added perturbations.

**Ablation Analyses:** Table 4 carries out ablation analysis on the working components of *Flareon*. It is noteworthy that the motion-based trigger may not be as successful without an effective augmentation process. Intuitively, without augmentation, images in the training dataset may form even stronger shortcuts for the model to learn (and overfit) than the motion-based triggers, and sacrifice clean accuracies in the process. Additionally, replacing the motion-based transform with uniformly-sampled pixel-wise triggers under the same $L^2$ distortion budget notably harms the resulting model's clean accuracy, adds visually perceptible noises, and can easily be detected with Grad-CAM (as

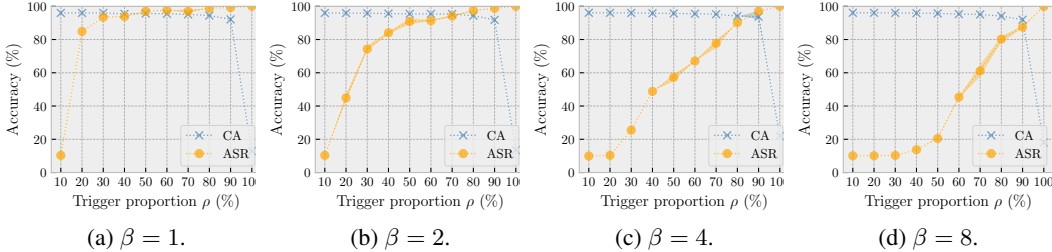

|     (a) $\beta = 1$.     |     (b) $\beta = 2$.     |     (c) $\beta = 4$.     |     (d) $\beta = 8$.     |

Figure 3: Effect of varying trigger initialization $\beta \in \{1, 2, 4, 8\}$ and $\rho \in [10\%, 100\%]$ for constant triggers. The trigger ratio $\rho$ provides a mechanism to tune the trade-off between CA and ASR, and lower $\beta$ improves ASR, but with increasing perturbation scales. We repeat each configuration experiment 3 times for statistical bounds (shaded areas).

Table 2: Comparing the noise added ($L^2$ distances from natural images) by constant triggers and their respective clean accuracies (%) and attack success rates (%).

| Datasets | CIFAR-10 | | | | CelebA | | | *tiny*-ImageNet | |
|---|---|---|---|---|---|---|---|---|---|
| Trigger initialization ($\beta$) | 1 | 2 | 4 | 8 | 2 | 4 | 8 | 1 | 2 |
| $L^2$ distance | 1.99 | 1.65 | 1.27 | 0.92 | 2.63 | 1.96 | 1.42 | 6.35 | 4.53 |
| Clean accuracy (%) | 94.49 | 94.43 | 94.11 | 94.10 | 80.11 | 79.87 | 79.69 | 57.14 | 57.23 |
| Attack success rate (%) | 98.82 | 97.88 | 90.08 | 82.51 | 99.88 | 99.16 | 99.89 | 98.44 | 74.23 |

Table 3: Comparing the noise added ($L^2$ distances from natural images) by learned triggers and their respective clean accuracies (%) and attack success rates (%).

| Datasets | CIFAR-10 | | | CelebA | | *tiny*-ImageNet | |
|---|---|---|---|---|---|---|---|
| Learned trigger bound ($\epsilon$) | 0.3 | 0.2 | 0.1 | 0.02 | 0.01 | 0.3 | 0.2 |
| $L^2$ distance | 0.88 | 0.67 | 0.37 | 0.19 | 0.11 | 2.14 | 1.40 |
| Clean accuracy (%) | 95.34 | 95.15 | 95.10 | 77.91 | 78.20 | 54.62 | 55.42 |
| Attack success rate (%) | 94.31 | 91.76 | 84.23 | 99.92 | 99.40 | 82.15 | 79.14 |

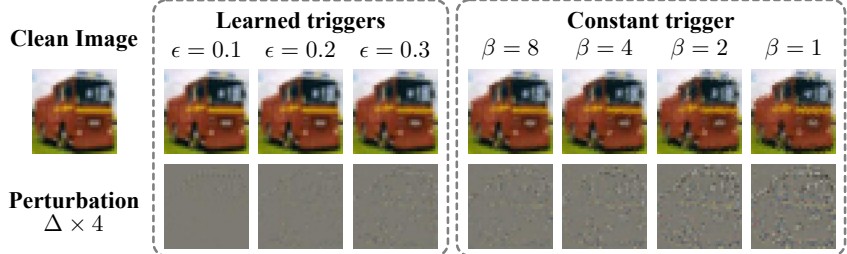

Figure 4: Visualizations of test-time perturbation noises (amplified $4\times$ for clarity) on CIFAR-10. Note that with larger $\beta$ values, the motion-based noise added to the original image becomes increasingly visible, whereas learned variants can notably reduce noise introduced by the trigger, while preserving high ASRs. For numerical comparisons, refer to Table 2 and Table 3.

shown in Figure 9 in the appendix). Appendix D also provides ablation results that compare trigger initializations from Uniform, Gaussian and Beta distribution samples.

**Trainer-Controlled Environments:** The design of *Flareon* does not assume any prior knowledge on the model architecture and training hyperparameters, making it a versatile attack on a wide variety of training environments. To empirically verify its effectiveness, we carry out CIFAR-10 experiments on different model architectures, namely ResNet-50 [14], squeeze-and-excitation networks with 18 layers (SENet-18 [15]), and MobileNet V2 [35]. Results in Table 5 show high ASRs with minimal degradation in CAs when compared against SGD-trained baselines. Table 6 presents additional results for CelebA and *tiny*-ImageNet that shows *Flareon* is effective across datasets and transform proportions $\rho$. Finally, Figure 6 in the appendix shows that *Flareon* can preserve the backdoor ASRs with varying batch sizes and learning rates.

**Defense Experiments:** As *Flareon* conceals itself within the data augmentation pipeline, it presents a challenge for train-time inspection to detect. Alternative defense thus involves backdoor removal methods. We thus further investigate its performance against these defenses including ABL [22], NAD [23], I-BAU [46], and ANP [43]. Due to page limits, additional defenses (Randomized Smoothing [40], RAB [42], Neural Cleanse [41], Fine-Pruning [25], Grad-CAM [36], and STRIP [11]) can be found in Appendix E. We conducted experiments on CIFAR-10 to evaluate these defense methods following default settings of the respective papers. Table 7 shows that the backdoor removal methods cannot decrease ASRs without degrading CAs first on *Flareon*-trained models. As ANP modifies the model, we trained models with ANP under the *Flareon* attack, and report its baseline separately. We believe that the reason for *Flareon*'s ability to persist even after these strong backdoor removal techniques due to our threat model allows for much higher trigger proportions than previously explored by these mainstream defenses.

Table 4: Ablation analysis of *Flareon* under $\beta = 2$.

| Ablation | Baseline | CA | ASR |
|---|---|---|---|
| No augmentation | 92.26 | 78.23 | 65.91 |
| RandAugment [5] | 96.14 | 95.35 | 94.12 |
| AutoAugment [4] | 96.05 | 95.16 | 97.01 |
| Pixel-wise triggers | 96.14 | 88.27 | 99.42 |

Table 5: Reliability across architecture choices.

| Model | Baseline | CA | ASR |
|---|---|---|---|
| PreActResNet-18 [14] | 95.66 | 94.38 | 97.79 |
| ResNet-50 [14] | 96.04 | 95.83 | 94.15 |
| SENet-18 [15] | 95.37 | 95.12 | 94.35 |
| MobileNet V2 [35] | 95.34 | 94.59 | 97.28 |
| DenseNet-121 [16] | 95.91 | 95.04 | 90.70 |

Table 6: Robustness against dataset choices. CA and ASR values are all percentages (%). Varying test-time stealthiness $\beta$ and transform proportion $\rho$ for constant triggers. The baseline clean accuracies without performing attacks are 78.92% for CelebA, and 58.84% for *tiny*-ImageNet.

| Datasets | CelebA ($\beta = 1$) | | CelebA ($\beta = 2$) | | *tiny*-IN ($\beta = 1$) | | *tiny*-IN ($\beta = 2$) | |
|---|---|---|---|---|---|---|---|---|
| $\rho =$ | CA | ASR | CA | ASR | CA | ASR | CA | ASR |
| 70% | 79.13 | 99.85 | 78.87 | 99.41 | 57.85 | 94.72 | 57.76 | 43.75 |
| 80% | 78.88 | 99.98 | 80.11 | 99.88 | 57.14 | 98.44 | 57.23 | 74.23 |
| 90% | 76.84 | 99.94 | 77.62 | 99.91 | 54.05 | 99.72 | 55.06 | 96.57 |

Table 7: The evaluations of *Flareon* on defenses. Note that all defenses assumes a small amount of clean data for backdoor removal. "Before" shows the baseline models produced by *Flareon*-poisoned training. "Clean" assumes a fresh un-poisoned fine-tuning environment. "Poisoned $p\%$" means the fine-tuning environment is also poisoned with *Flareon*, and it can adjust $p\%$ to stochastically transforms $p\%$ of the clean data with triggers. "—" means it is not applicable to *Flareon*.

| Method | Before | | Clean | | Poisoned (20%) | | Poisoned (50%) | |
|---|---|---|---|---|---|---|---|---|
| | CA (%) | ASR (%) | CA (%) | ASR (%) | CA (%) | ASR (%) | CA (%) | ASR (%) |
| ABL [22] | | | 56.97 | 99.76 | — | — | — | — |
| NAD [23] | 94.43 | 97.88 | 36.56 | 15.22 | 18.63 | 47.59 | 10.41 | 76.32 |
| I-BAU [46] | | | 10.22 | 92.30 | 9.96 | 97.66 | 10.68 | 98.52 |
| ANP [43] | 89.37 | 97.46 | 83.52 | 17.96 | 82.34 | 98.50 | 81.16 | 99.74 |

Table 8: The evaluations of WaNet and BadNets to enable *all2all* attacks on CIFAR-10. Note that "$\rho$" denotes the stochastic trigger proportion, "—" means it is not applicable to *Flareon*.

| Method | $\rho$ (%) | BadNets [13] | | WaNet [29] | | *Flareon* | |
|---|---|---|---|---|---|---|---|
| | | CA (%) | ASR (%) | CA (%) | ASR (%) | CA (%) | ASR (%) |
| Clean Label | 50 | 42.69 | 80.52 | 81.34 | 83.79 | 95.39 | 90.89 |
| | 80 | 21.88 | 99.93 | 91.24 | 63.87 | 94.43 | 97.88 |
| Dirty Label | 10 | 66.19 | 99.73 | 90.33 | 78.94 | — | — |
| | 20 | 59.97 | 99.85 | 88.95 | 87.24 | — | — |

**Comparisons to Extensions of Existing Attacks:** We extend WaNet [29] and BadNets [13] as shown in Table 8 ("Clean Label") to further enable *all2all* attacks under clean-label training. Note that they also introduce much larger distortions (as shown in Figure 5). We highlight that without the joint learning of backdoor and normal training objectives, they performed substantially worse in terms of the trade-offs between clean accuracies and attack success rates. In "Dirty Label" rows, we further allow both baselines to use label changes. For a small ratio of all sampled images, we randomly replace the label with a different random target and apply the corresponding triggers, in order to learn backdoors. Note that it clearly fails to meet the train-time stealthiness requirements, as not only are the triggers easy to identify, but they also modify training labels.

## 6 CONCLUSION

This work presents *Flareon*, a simple, stealthy, mostly-free, and yet effective backdoor attack that specifically targets the data augmentation pipeline. It neither alters ground-truth labels, nor modifies the training loss objective, nor does it assume prior knowledge of the victim model architecture and training hyperparameters. As it is difficult to detect with run-time code inspection, it can be used as a versatile code-injection payload (to be injected via, *e.g.*, dependency confusion, name-squatting, or feature proposals) that disguises itself as a powerful data augmentation pipeline. It can even produce models that learn target-conditioned (or "*all2all*") backdoors. Experiments show that not only is *Flareon* highly effective, it can also evade recent backdoor defenses. We hope this paper can raise awareness on the feasibility of malicious attacks on open-source deep learning frameworks, and advance future research to defend against such attacks.

## 7 REPRODUCIBILITY STATEMENT

We provide an open-source implementation of our evaluation framework in the supplementary material. All experiments in the paper uses public datasets, *e.g.*, CIFAR-10, CelebA, *tiny*-ImageNet. Following the README file, users can run *Flareon* experiments on their own device to reproduce the results shown in paper with the hyperparameters in Appendix B.

## 8 ETHICS STATEMENT

We are aware that the method proposed in this paper may have the potential to be used by a malicious party. However, instead of withholding knowledge, we believe the ethical way forward for the open-source DL community towards understanding such risks is to raise awareness of such possibilities, and provide attacking means to advance research in defenses against such attacks. Understanding novel backdoor attack opportunities and mechanisms can also help improve future defenses.

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

# A  A PAYLOAD DELIVERY PROOF-OF-CONCEPT EXAMPLE WITH A PYTORCH EXPLOIT

## A.1  TRIGGER CONSENSUS BETWEEN ATTACKER AND VICTIM

To enforce consensus between the attacker and the victim on the constant triggers to use, we propose the following approach in Algorithm 2. Here, $\beta$ is the Beta distribution parameter, $(H, W)$ is the image resolution, and $s$ is the random seed. For each target label $t$, we sample a Beta distribution with random seed $s$, and normalize the sampled values to $[-1, 1]$. It then updates the random seed for the next trigger sampling using a pseudorandom number generator (PRNG), for instance, using a secure hash function SHA-256. By generating the triggers sequentially for each class, we can ensure that the attacker and the victim will use the same set of triggers regardless the number of classes.

---

**Algorithm 2** The constant trigger generation algorithm shared between victim and attacker.

**function** GenerateTrigger($\beta, (H, W), s$)
    $\tau \leftarrow 0$          ▷ Initialize triggers for all classes.
    **for** $t \in \mathcal{C}$ **do**          ▷ For each target label. . .
        $\mathbf{b} \sim \mathcal{B}_{s,\beta,\beta}(H, W, 2)$      ▷ . . . sample the Beta distribution for triggers using random seed $s$.
        $\tau_t \leftarrow 2\mathbf{b} - 1$      ▷ Normalize to $[-1, 1]$, and assign to the $t^{\text{th}}$ trigger.
        $s \leftarrow \text{PRNG}(s)$      ▷ Change seed for the next trigger sampling with a PRNG.
    **end for**
    **return** $\tau$
**end function**

---

## A.2  EXAMPLE TRIGGER DELIVERY

We designed a proof-of-concept (PoC) payload delivery[3]. It compiles a malicious PyTorch checkpoint of a pretrained ResNet-18 model from Torchvision [31]. If the checkpoint is loaded using `torch.load`, it will replace the `torch.util.data.dataloader` module source file with a variant that contains the *Flareon* payload. As a result, future classifier models trained in this Python environment will always include *Flareon* backdoors.

This example follows Appendix A.1 to ensure consensus between the training process and the attacker on the constant triggers to use. If it is to be extended for the learned variant, it would be necessary to deliver the learned triggers. However, the trigger delivery mechanism can also be code-injected along with the attack. For instance, the attacker may choose to upload the triggers upon training completion or conceal the triggers in the model checkpoint. Such an attack does not violate the "no prior knowledge" assumption in Table 1, since the attacker can deploy the payload even before the victim appears.

## A.3  TRIGGER-TARGET PAIR RECOVERY ON DEPLOYED MODELS

Let's assume the attacker is given access to providing input and reading the output of a deployed model that has been poisoned, for instance, through an API. In that case, we propose the following approach illustrated by Algorithm 3 to recover all pairs of triggers and their associated labels effortlessly.

First, we can use any clean image and transform it using all possible triggers. Next, we observe the model output for the transformed images. It is important to note that the Flareon-poisoned models are learned to provide higher priority to all triggers than clean image features. Therefore, with high probability, each trigger corresponds to a unique target output. Finally, suppose the poisoned model contains only $|\mathcal{C}|$ class outputs. In that case, any triggers sampled beyond the $|\mathcal{C}|$-th one will have no effect on the model, as the additional triggers are not used during training and will not be associated with any class. Consequently, we expect the output to correspond to the original label of the clean image.

---

[3]The source code of the PoC example can be found in the supplementary material, along with a pre-compiled checkpoint. For safety precautions, please refrain from loading the checkpoint.

This approach allows us to identify which trigger has been associated with the image class, making it feasible to recover all pairs of trigger and their respective labels. In our tests with ResNet-18 models trained on CIFAR-10 under *Flareon*, we recovered all 10 pairs of triggers and their associated labels with $100\%$ accuracy.

---

**Algorithm 3** The algorithm for target-trigger pair recovery.

---

**function** RecoverTargetTrigger($\mathcal{D}_{\text{rec}}, f$)
    $M \leftarrow \{\}$
    **for** $\tau$ in $\boldsymbol{\tau}$ **do**
        $\mathbf{x} \leftarrow \mathcal{T}_\tau(\mathcal{D}_{\text{rec}})$                       $\triangleright$ Apply the current trigger $\tau$ to the images in the dataset.
        $\mathbf{n} \leftarrow f(\mathbf{x})$                          $\triangleright$ Retrieve class names of the images with the API $f$.
        $\mathbf{c} \leftarrow \text{bincount}(\mathbf{n})$                  $\triangleright$ Count the number of name occurrences.
        $n \leftarrow \arg\max(\mathbf{c})$                 $\triangleright$ Find the most frequent name by majority vote.
        **if** $\mathbf{c}_n > \text{sum}(\mathbf{c})/2$ **then**
            $M_n \leftarrow \tau$             $\triangleright$ Assign the trigger to the class name on majority consensus.
        **else**
            $M_n \leftarrow \text{null}$       $\triangleright$ Otherwise assign null as the trigger $\tau$ is likely unused by $f$.
        **end if**
    **end for**
    **return** $M$
**end function**

---

# B  EXPERIMENTAL SETUP

## B.1  DATASETS

**CIFAR-10** consists of 60,000 $32 \times 32$ resolution images, of which 50,000 images are the training set and 10,000 are the test set. This dataset contains 10 classes, each with 6000 images [17].

**CelebA** is a large face dataset containing 10,177 identities with 202,599 face images. Following previous work [33], we select three balanced attributes from the 40 attributes: heavy makeup, mouth slightly, and smile, and combine the three attributes into 8 classes. For training, the baseline uses no augmentations on the images.

**Tiny-ImageNet** is an image classification dataset containing 200 categories, each category with 500 training images, 50 validation and 50 test images [21]. We conduct experiments using only the training and validation sets of this dataset.

Table 9 shows the details of these datasets.

Table 9: Overview of the datasets used in this paper.

| Dataset | ‖ | Input size | Train-set | Test-set | Classes |
|---|---|---|---|---|---|
| CIFAR-10 | ‖ | $32 \times 32 \times 3$ | 50,000 | 10,000 | 10 |
| CelebA | ‖ | $64 \times 64 \times 3$ | 162,770 | 19,962 | 8 |
| *tiny*-ImageNet | ‖ | $64 \times 64 \times 3$ | 100,000 | 10,000 | 200 |

## B.2  MODELS AND HYPERPARAMETERS

We evaluate *Flareon* using ResNet-18, MobileNet-v2, and SENet-18. The optimizer for all experiments uses SGD with a momentum of 0.9. Tables 10 and 11 provides the default hyperparameters used to train *Flareon* models.

# C  TRIGGER VISUALIZATIONS

In this section, we show the visualization of triggers on CelebA and *tiny*-ImageNet. Figure 5 show the clean samples and the samples after applying the motion-based triggers.

Table 10: Default hyperparameters for constant *Flareon* triggers.

| Dataset | | CIFAR-10 | CelebA | *tiny*-ImageNet |
|---|---|---|---|---|
| Model learning rate $\alpha_{\text{model}}$ | | 0.01 | 0.01 | 0.01 |
| Model learning rate decay | | 1/2 every 30 epochs | None | 1/2 every 30 epochs |
| Weight decay | | 5e-4 | 5e-4 | 5e-4 |
| Epochs | | 350 | 50 | 400 |
| Batch size | | 128 | 128 | 128 |

Table 11: Default hyperparameters for adaptive *Flareon* triggers.

| Dataset | | CIFAR-10 | CelebA | *tiny*-ImageNet |
|---|---|---|---|---|
| Model learning rate $\alpha_{\text{model}}$ | | 0.01 | 0.01 | 0.01 |
| Model learning rate decay | | 1/2 every 30 epochs | None | 1/2 every 30 epochs |
| Trigger learning rate $\alpha_{\text{flareon}}$ | | 0.2 | 0.2 | 0.2 |
| Weight decay | | 5e-4 | 5e-4 | 5e-4 |
| Epochs | | 400 | 80 | 600 |
| Batch size | | 128 | 128 | 128 |

## D    ADDITIONAL RESULTS

Figure 6 shows that *Flareon* can preserve the backdoor ASRs with varying batch sizes and learning rates. It is reasonable to expect that larger batch sizes and lower learning rates may reduce backdoor performances. Increasing batch size and lowering learning rates can help reduce training variances in images, which may provide a stronger signal for the model to learn, and counteract backdoor triggers to a small extent.

We additionally compare the use of Uniform $\mathcal{U}(-s, s)$, Beta $\mathcal{B}(\beta, \beta)$, and Gaussian $\mathcal{N}(0, \sigma)$ initialized triggers in Table 12. Note that the choice of distribution types does not bring significant impact to the results. The rationale of choosing a Beta distribution is because it is nicely bounded within $[-1, 1]$, effectively limiting the perturbation of each pixel to be within its immediate neighbors. Besides, Beta distributions encompass Uniform distribution, *i.e.*, $\mathcal{B}(\beta, \beta)$ is Uniform when $\beta = 1$. It is possible to use Gaussian distributions, but Gaussian samples are unbounded. Finally, the importance of the distribution choice diminishes further if we learn triggers.

We visualize the confusion matrix and ASR matrix of the *Flareon*-trained CIFAR-10 model. The confusion matrix in Figure 7a shows that *Flareon* does not noticeably impact clean accuracies of all labels. Moreover, the ASR matrix in Figure 7b further shows the capabilities of *all2all* backdoors. Namely, any images of *any* class can be attacked with *any* target-conditioned triggers with very high success rates.

Table 12: Ablation on different distribution choices (Uniform $\mathcal{U}(-s, s)$, Beta $\mathcal{B}(\beta, \beta)$, and Gaussian $\mathcal{N}(0, \sigma)$) on the trigger initialization of *Flareon* on CIFAR-10, sorted by $L^2$ distances in ascending order. Note that Beta $\mathcal{B}(1, 1)$ is equivalent to the Uniform sampling within $[-1, 1]$. Beta distribution with $\beta = 2$ has better ASR with lower $L^2$ changes. The importance of initialization diminishes if we learn triggers. We rerun each setting 5 times with different seeds for statistical bounds.

| Distribution | $L^2$ distance ($\downarrow$) | Clean accuracy (%) | Attack success rate (%) |
|---|---|---|---|
| Uniform ($s = 0.70$) | $1.50 \pm 0.05$ | $94.51 \pm 0.32$ | $92.66 \pm 0.52$ |
| Uniform ($s = 0.75$) | $1.61 \pm 0.07$ | $94.22 \pm 0.12$ | $93.74 \pm 0.66$ |
| Beta ($\beta = 2$) | $1.67 \pm 0.07$ | $94.29 \pm 0.14$ | $97.25 \pm 0.63$ |
| Uniform ($s = 0.8$) | $1.77 \pm 0.09$ | $94.21 \pm 0.22$ | $95.51 \pm 1.04$ |
| Gaussian ($\sigma = 0.5$) | $1.84 \pm 0.06$ | $94.73 \pm 0.09$ | $91.24 \pm 2.13$ |
| Beta ($\beta = 1$) | $2.04 \pm 0.12$ | $94.41 \pm 0.08$ | $98.80 \pm 0.07$ |
| Gaussian ($\sigma = 0.75$) | $2.74 \pm 0.11$ | $94.13 \pm 0.14$ | $95.17 \pm 0.76$ |

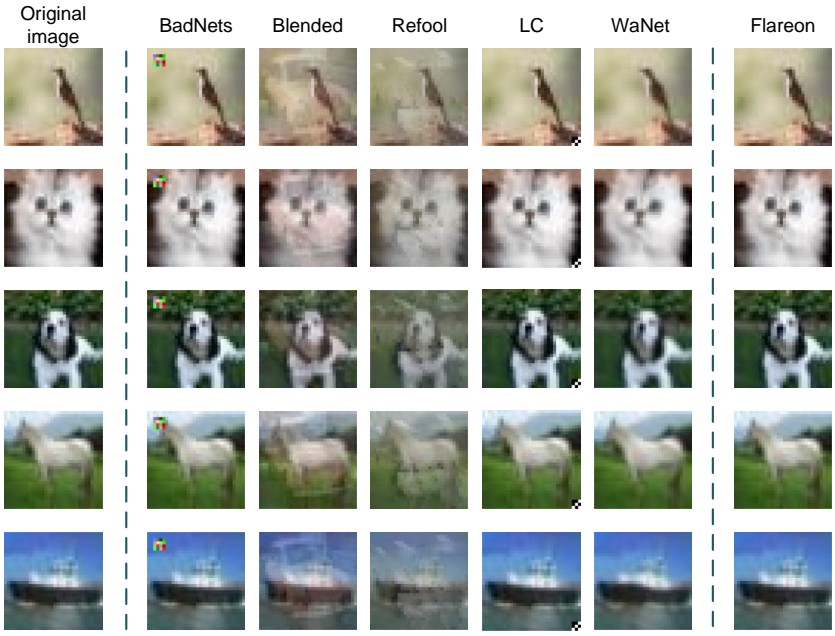

Figure 5: Comparing the test-time triggers of recent backdoor attacks (Patched [13], Blended [3], Refool [27], LC [38], and WaNet [29]).

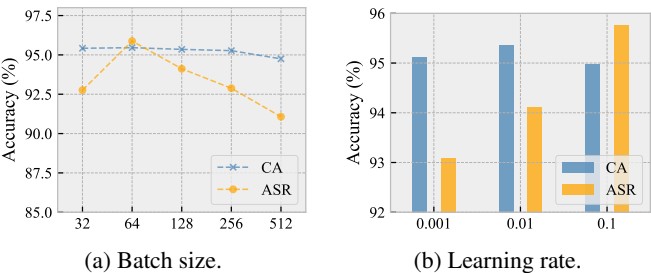

(a) Batch size.                    (b) Learning rate.

Figure 6: Varying batch sizes and learning rates.

## E    ADDITIONAL DEFENSE EXPERIMENTS

**Fine-Pruning:** In Figures 8a to 8c, we test Fine-Pruning [25] on the *Flareon*-backdoored models, and find backdoor neurons persist well against fine-pruning, as CAs can degrade at a faster rate than ASRs *w.r.t.* increasing channel sparsity.

**STRIP:** Figures 8d to 8f shows that the entropy distribution of *Flareon* models is similar to that of the clean model.

**Neural Cleanse:** Figure 8g shows that neural cleanse is unable to detect backdoors generated by *Flareon* with constant triggers. With adaptive trigger learning, learned triggers with smaller perturbations are, however, showing higher anomaly (Figure 8h). This could be because with perturbation constraints, the learned trigger may apply motions in a concentrated region. While it is possible to introduce NC evasion loss objective [1] to avoid detection, it incurs additional overhead in model forward/backward passes. To defend against NC with *Flareon*, it is thus best to adopt randomly initialized constant triggers.

**Grad-CAM:** Visualization tools such as Grad-CAM [36] are helpful in providing visual explanations of neural networks. Following [29], we also evaluate the behavior of backdoored models against such tools. Pixel-wise triggers as used in Table 4 are easily exposed due to its fixed trigger pattern (Figure 9).

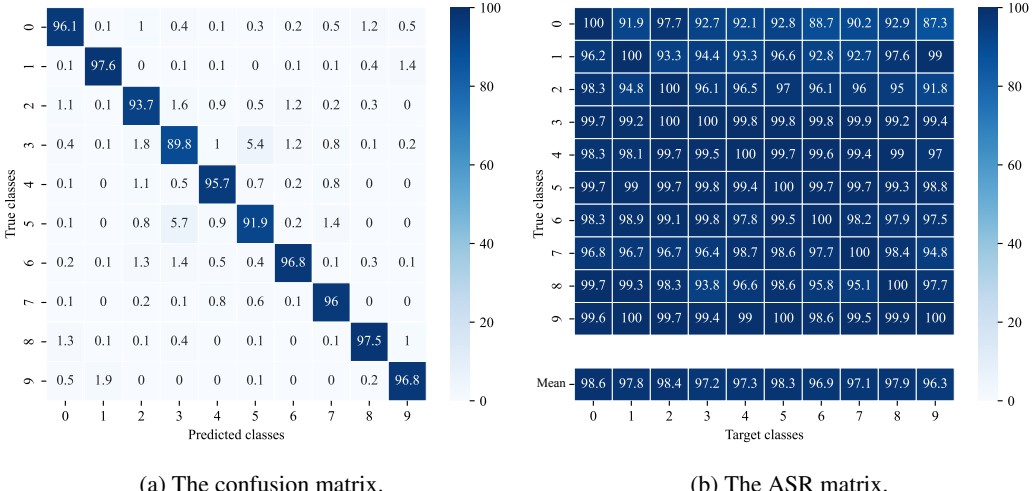

(a) The confusion matrix.       (b) The ASR matrix.

Figure 7: Class-wise statistics for the CIFAR-10 model. (a) The confusion matrix between the model prediction and ground-truth classes. (b) The ASR matrix shows the ASR values of attacking all test images of any label with any target class. "Mean" reports the overall ASR of each target.

**Smoothing-based Defenses:** To demonstrate the reliability of *Flareon*, we apply Randomized Smoothing [40] on *Flareon* with different trigger proportions $\rho$, as shown in the Table 13. In addition, we follow the setup of RAB [42], an ensemble-based randomized smoothing defense, and use the official implementation for empirical robustness evaluation, which sets the number of sampled noise vectors to $N = 1000$, and samples the smoothing noise from the Gaussian distribution $\mathcal{N}(0, 0.2)$ on CIFAR-10. For fairness, we use the same CNN model and evaluation methodology in RAB. The experimental results are in Table 14. *Flareon* enjoys great success under smoothing-based defenses.

Table 13: Evaluation of randomized smoothing on *Flareon*.

| Model | $\rho =$ | 50 | 60 | 70 | 80 | 90 |
|---|---|---|---|---|---|---|
| CIFAR-10 | Clean accuracy (%) | 92.24 | 87.82 | 85.38 | 76.37 | 63.72 |
| | Attack success rate (%) | 97.33 | 96.70 | 98.10 | 99.42 | 99.16 |

Table 14: Evaluation of RAB on *Flareon*. "Vanilla" denotes training without RAB. Following [42] for evaluation, the empirical robust accuracy reports the proportion of malicious inputs that not only attacks the vanilla model successfully, but also tricks RAB.

| Model | Benign Accuracy (%) | | Empirical Robust Accuracy under *Flareon* (%) | | | | |
|---|---|---|---|---|---|---|---|
| | Vanilla | RAB | Vanilla | $\rho = 50\%$ | $\rho = 60\%$ | $\rho = 70\%$ | $\rho = 80\%$ |
| CIFAR-10 | 61.71 | 58.74 | 0 | 9.71 | 8.15 | 6.45 | 3.82 |

## F  VISUALIZATION

In this section, we visualize the training-phase augmentations introduced by *Flareon* on CIFAR-10, CelebA and *tiny*-ImageNet. Figure 10 shows the clean samples, the samples after RandAugment and AutoAugment, and samples after applying triggers. Figure 13 shows the clean samples and the samples transformed with motion-based triggers of different intensities.

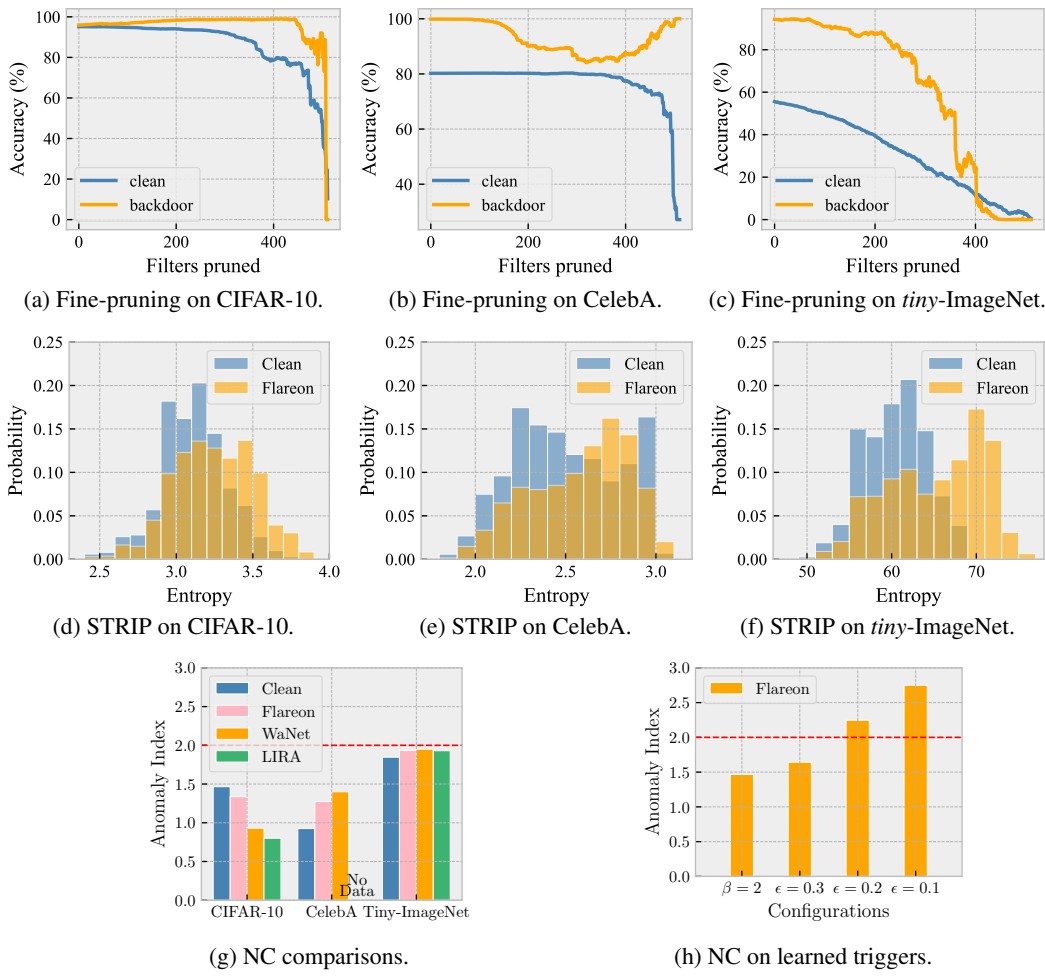

Figure 8: (a, b, c) Fine-pruning for the CIFAR-10, CelebA and *tiny*-ImageNet models. (d, e, f) STRIP defenses on the CIFAR-10, CelebA and *tiny*-ImageNet models. (g) *Flareon* evades detection by Neural Cleanse (NC) on the CIFAR-10, CelebA and *tiny*-ImageNet models. (h) Smaller perturbations are easier to detect for Neural Cleanse (NC) on CIFAR-10.

## G   COMPUTATIONAL RESOURCES

*Flareon* uses up to 0.8, 0.08, or 2.2 GPU-hours per experiment run for CIFAR-10, CelebA, and *tiny*-ImageNet datasets respectively on NVIDIA Tesla V100 GPUs. We estimate a total of 370 GPU-hours for all experiments presented in this paper.

## H   ETHICS STATEMENT & LIMITATIONS

We are aware that the method proposed in this paper may have the potential to be used by a malicious party. However, instead of withholding knowledge, we believe the ethical way forward for the open-source DL community towards understanding such risks is to raise awareness of such possibilities, and provide open-source attacking means to advance research in defenses against such attacks. Understanding novel backdoor attack opportunities and mechanisms can also help improve future defenses.

We demonstrate the effectiveness of *Flareon* in datasets with up to $|\mathcal{C}| = 200$ classes (*tiny*-ImageNet). As the number of backdoors $|\mathcal{C}|^2$ increase quadratically *w.r.t.* the number of classes, it may not scale to models with a huge number of classes. We intend to study this in the future with larger scale

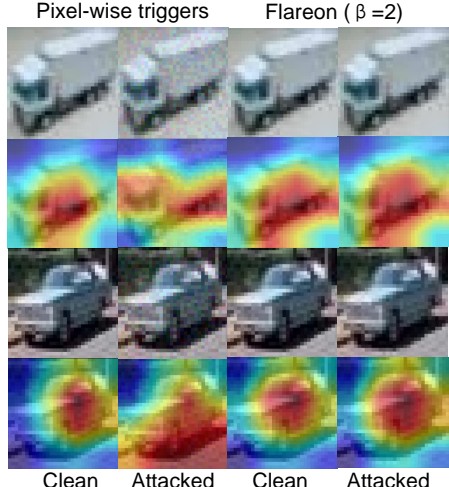

Figure 9: Grad-CAM heat maps and perturbed images comparisons between *Flareon* and pixel-wise triggers.

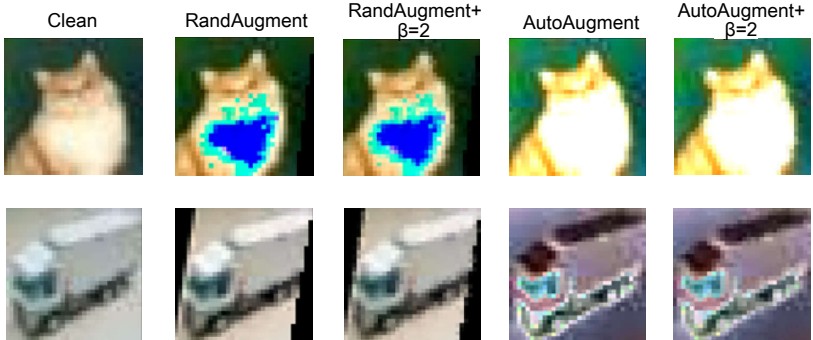

Figure 10: Visualizations of the *Flareon* triggers on the CIFAR-10 dataset. "$+$" represents the application of a motion-based trigger with $\beta = 2$.

datasets. We also note that similar to other attacks, currently *Flareon* can only backdoor image classifier models. It is possible to extend this attack to other image tasks and is a potential direction for future work.

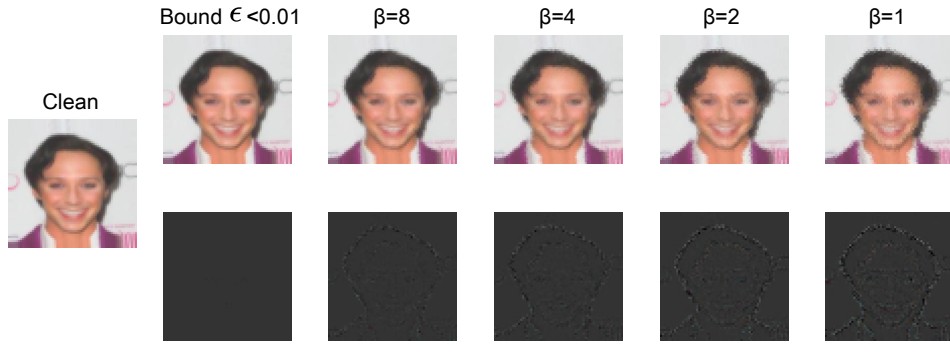

Figure 11: Visualizations of the *Flareon* triggers on the CelebA dataset.

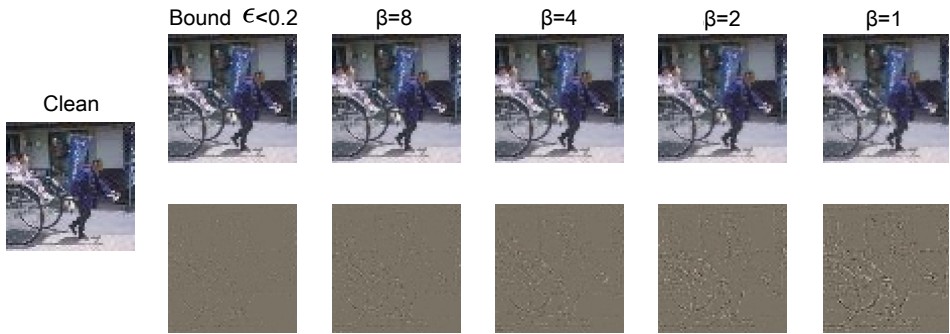

Figure 12: Visualizations of the *Flareon* triggers on the *tiny*-ImageNet dataset.

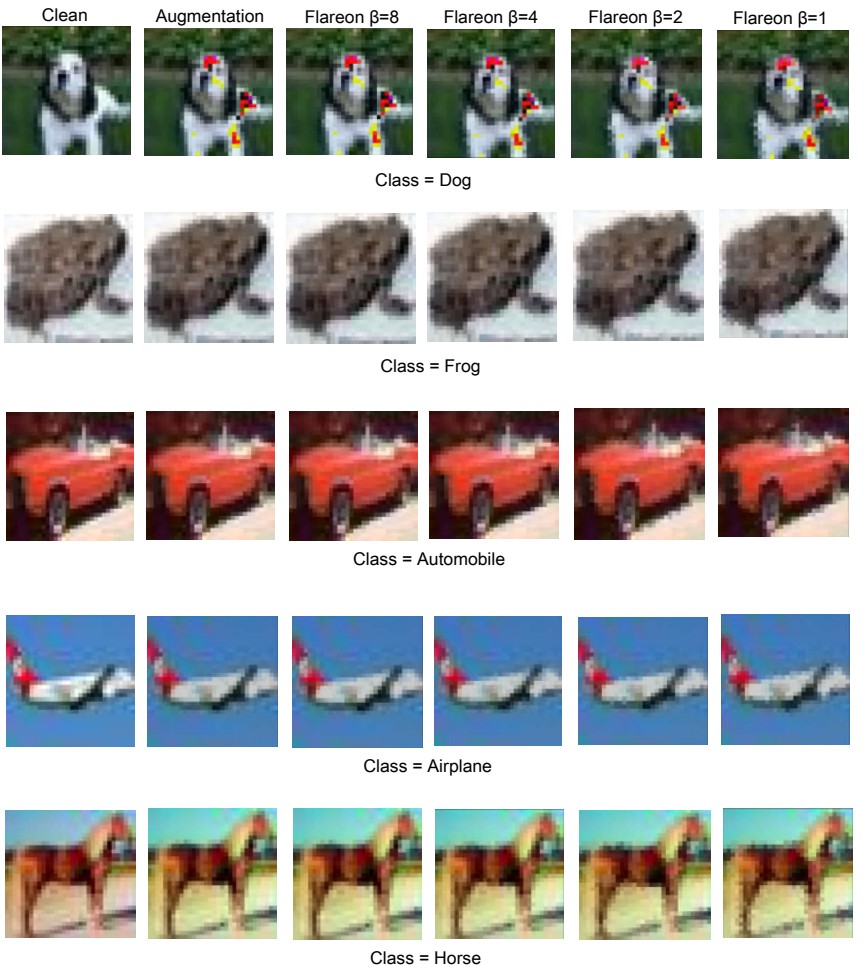

Figure 13: Visualizations of the *Flareon* triggers on the CIFAR-10 dataset. It includes the clean samples, augmentation samples and the Flareon attacks with $\beta \in \{8, 4, 2, 1\}$ respectively.

# I LICENSES

Table 15 lists the relevant resources used in this paper, and their respective licenses. Note that the resources are all publicly available and open-source, and widely used in the machine learning community, but some provided no licenses.

Table 15: Open-source resources used in this paper.

| Name | License | URL |
|---|---|---|
| PyTorch | BSD | GitHub: pytorch/pytorch |
| **Methods** | | |
| WaNet [29] | GPL 3.0 | GitHub: VinAIResearch/Warping-based_Backdoor_Attack-release |
| Neural Cleanse [41] | MIT | GitHub: bolunwang/backdoor |
| I-BAU [46] | MIT | GitHub: YiZeng623/I-BAU |
| Fine-Pruning [25] | — | GitHub: kangliucn/Fine-pruning-defense |
| STRIP [11] | — | GitHub: garrisongys/STRIP |
| RAB [42] | — | GitHub: AI-secure/Robustness-Against-Backdoor-Attacks |
| ABL [22] | — | GitHub: bboylyg/ABL |
| ANP [43] | — | GitHub: csdongxian/ANP_backdoor |
| **Datasets** | | |
| CIFAR-10 [17] | — | `https://www.cs.toronto.edu/~kriz/cifar.html` |
| CelebA [28] | — | `http://mmlab.ie.cuhk.edu.hk/projects/CelebA.html` |
| *tiny*-ImageNet [21] | — | `http://cs231n.stanford.edu/tiny-imagenet-200.zip` |

