# OpenReview forum: "Invisible and Adaptive Training-Phase Target-Conditioned Backdoors"
_ICLR.cc/2024/Conference — ICLR 2024 Conference Withdrawn Submission_

### Official Review · Reviewer_uTzJ · 2023-10-17

**Soundness:** 2 fair
**Presentation:** 2 fair
**Contribution:** 2 fair
**Rating:** 5
**Confidence:** 4

**Summary:**

In this work, the authors introduce a new type of backdoor attack that implants backdoors to models by injecting malicious code payload into the data augmentation pipeline. The authors propose a motion-based trigger generation method that can bypass various defenses. Extensive experiments and ablation studies are conducted to show that the attack can achieve decent clean accuracy and attack success rate as well as circumvent multiple defenses.

**Strengths:**

1. The authors propose a new threat model that has never been studied before and the proposed attack possesses good attributes such as sample-specific, clean-label and learnable triggers.

2. The authors conduct abundant experiments and extensive ablation studies, evaluating the attack from different perspectives.

3. The proposed attack can achieve decent CA and ASR and bypass several existing defenses. The proposed adaptive trigger further improves the stealthiness of the attack.

**Weaknesses:**

1. A primary concern is that the threat model looks too strong and unrealistic. Injecting malicious code payload into the data augmentation pipeline is easy to detect upon human inspection since data augmentation is actually an important segment of model training and is not often neglected in code development.

2. The authors claim that the attacker doesn't have a control of the training process. However, there are a few new hyper-parameters introduced in Algorithm 1, are these parameters pre-defined? If so, the flexibility of the attack would be low and the performance and stealthiness may not be guaranteed, if not, how to select/decide optimal hyper-parameters without interfering the original training process?

3. Advanced training for CNN-based and Transformer-based models usually requires complex and strong data augmentations (e.g., a mixture of RandAug, CutOut, MixUp, CutMix, etc.)  rather than a single, easy data augmentation. As the proposed attack requires modification of the data augmentation pipeline, it is necessary to show how different data augmentation would affect the performance of the attack. I noticed that CutOut and RandAug are used in the paper evaluation, what about other mainstream data augmentations such as CutMix and MixUp?

4. It would be desirable to compare the attack performance to more advanced attacks such as LIRA, Sleeper Agent, Marksman, Input-aware dynamic backdoor [1]  and Invisible Backdoor Attack with Sample-Specific Triggers [2]. [1] and [2] use sample-specific triggers that are similar to the proposed attack.

[1]. Input-Aware Dynamic Backdoor Attack, Tuan Anh Nguyen et. al, NeurIPS 2020.

[2]. Invisible Backdoor Attack With Sample-Specific Triggers, Yuezun Li et. al, ICCV 2021.

**Questions:**

1. In Table 8, why is the CA of BadNets so low in both clean-label and dirty-label cases?

---

### Official Review · Reviewer_hF7C · 2023-10-26

**Soundness:** 3 good
**Presentation:** 2 fair
**Contribution:** 3 good
**Rating:** 5
**Confidence:** 3

**Summary:**

This paper introduces a code modification, referred to as *Flareon*, to data augmentation that introduces backdoors into the resultant model. This modification does not require any changes or put any conditions on any other part of the training process (e.g., optimizer, training data, loss, etc.). This work also proposes a threat model suited to this attack that assumes the victim is looking for some signs of compromise. Also, a novel way of modifying images by using ``motion''-based perturbations is described and used. The paper evaluates Flareon on a few standard datasets, showing attack success usually above 90%. The efficacy of the attack is tested against some prior existing defenses, across different DNN architectures, and a couple different augmentation strategies.

**Strengths:**

* (Major) The paper comprehensively examines Flareon's performance with different DNN architectures, datasets, data augmentations, and defenses.

* (Major) The results convincingly show an attack that usually has above 90% success rate in most conditions tested.

* (Moderate) The contributions of this paper are aided by included code for reproducibility.

* (Minor) The motivation for this work seems interesting, in that it shows how the prevalent use of machine-learning libraries could be a vulnerability, and that perhaps their integrity and trust should have more of a focus. I would weight this strength higher if the threat model were more precisely defined (discussed in weaknesses).

**Weaknesses:**

* (Major) One of the contributions is that this work introduces "a new code-injection threat model that requires the attack to satisfy restrictive train-time stealthiness specifications". However, I could not find a precise definition of what makes an attack "stealthy" or specifically why Flareon met that definition and prior work did not. As I understand it, Section 3 is meant to define the threat model, and it notes that the attacker's goal is to secretly introduce backdoors during training and should focus on stealthiness, but doesn't go into more detail than that. If another future work wants to make an attack for this threat model, it seems to me they would have a hard time determining if they fit within it. One way to help solve this could be to come up with measures that numerically quantify some of the aspects of "stealthiness" mentioned in other parts of the paper like lines of code-added, compute profiling differences, etc. Another place where this impreciseness is an issue is in Table 1, where some prior work is said to have "Test-time trigger stealthiness" and other prior work does not. I ask a question about this in Questions.

* (Moderate) According to the threat model in Section 3, the victim is assumed to be doing code inspection. The paper claims that this code would be hard to find with code inspection, but I did not see any specific justification or evidence for this. At the moment, it seems like it could be found by glancing at the augmentation code or noticing the inclusion of a large tensor the size of $2K$ input images that denote the hardcoded backdoor triggers.

* (Moderate) It is unclear how triggers that are learned can be retreived by the attacker, and it seems as though providing hard-coded triggers to bypass this may go against the stated threat model of being "stealthy" as it would require injecting a large tensor along with the code.

* (Minor) There are several points within some algorithms and equations that I found confusing (or are possibly typos) which makes it difficult to be sure of what the algorithm is doing.

**Questions:**

* In Table 1, some prior work is said to have "Test-time trigger stealthiness" and other prior work does not. How was this determined?

* In Table 4, what augmentation was pixel-wise triggers (instead of motion-based) paired with?

* Figure 2b shows new code that uses several variables not introduced, described, or obvious what they are (e.g., identity_grid, pert_grid, aa_images). Also, is it the case that data_augmentation modules are usually given the labels? Even in the "before" example the labels are simply returned without any use or modification. I would think, since augmentation is not meant to change the labels, that labels would not normally be included in a typical augmentation pipeline. Is that not the case? Could you point to currently used augmentation modules that do use the labels?

* In eq. 3, what is $\mathcal{L}^{\text{sce}}$? Is this a standard loss or some specific loss required for this objective?

* In eq. 5, it seems like $\tau$ has shape K \times H \times W \times 2, but the sentence directly before eq. 5 says that $\tau \in [-1,1]^{H \times W \times 2}$. Is it the case that $\tau_{i} \in [-1,1]^{H \times W \times 2}$?

* In alg. 1, where does $y$ come from on line 5? Is that supposed to be $\textbf{y}_i$?

* In Alg. 1, where does $\mathcal{T}$ come from or initialized? Is that defined and initialized outside of this algorithm?

* In Table 8, it seems like any poisoning of the fine-tuning environment violates the defense's assumptions of a clean dataset. Is that correct? If so, what is the takeaway from seeing the defenses not work well if they are not being used as intended?

* If the backdoor triggers are learned, how would an attacker retreive them and be able to use them? It looks like Appendix A.3 assumes the attacker knows the triggers used to be able to recover them, so it does not seem applicable to the learned triggers. If they are not learned, does Flareon's code injection also require an accompanied tensor of size $K \times W \times H \times 2$ to be included and hidden?

* If the victim is doing code inspection as assumed at the top of Section 3, then would they not catch this code injection? If not, why not?

---

### Official Review · Reviewer_jpj3 · 2023-10-28

**Soundness:** 3 good
**Presentation:** 2 fair
**Contribution:** 3 good
**Rating:** 5
**Confidence:** 4

**Summary:**

The paper proposes a training-phase backdoor attack called "Flareon" which targets the data augmentation pipeline with motion-based triggers. The proposed attack "Flareon" can be treated as a kind of code-injection threat because it simply modifies a small part of code to the data preprocessing pipeline. "Flareon" also can achieve all2all attacks, meaning that the triggers for all classes enjoy high success rates on all test set images. The experiments are conducted on three datasets and extensitive experiments show the effectiveness of proposed method.

**Strengths:**

The proposed backdoor attack is a more stealthy attack compared to most existing backdoor attacks, which can be proved in Table 1. Also, proposed attack is powerful target-conditioned attack and show high attack success rates (ASRs). Many ablation studies including trigger initialization, robustness against datasets and different defenses, are conducted to demonstrate the effectiveness of proposed attack.

**Weaknesses:**

Some concepts are not clear. For example, the paper shows that equation (3) can learn a shortcut between trigger patterns and target label. However, the paper does explain how to make the natural image difficult to learn.

In the paper, the concept of "all2all" attack is dfferent from the existing concept "all2all" attack [1]. However, there is no more description and experiments to demonstrate the difference between these two "all2all" settings. Also, there is another paper [2] that can attack any label. It is better to explain the difference between proposed method and Marksman backdoor [2].

[1] Nguyen, Tuan Anh, and Anh Tran. "Input-aware dynamic backdoor attack." Advances in Neural Information Processing Systems 33 (2020): 3454-3464.
[2] Doan, Khoa D., Yingjie Lao, and Ping Li. "Marksman backdoor: Backdoor attacks with arbitrary target class." Advances in Neural Information Processing Systems 35 (2022): 38260-38273.

**Questions:**

For the poison rate \rho, the experiments show that \rho needs to be 70% or 80% (e.g. in Table 6) to achieve high attack success rates (ASRs). In the case of high poison rate, can the proposed backdoor attack stealthy?

---

### Official Review · Reviewer_vAeD · 2023-11-01

**Soundness:** 2 fair
**Presentation:** 4 excellent
**Contribution:** 2 fair
**Rating:** 3
**Confidence:** 5

**Summary:**

The authors propose a code-injection backdoor attack named Flareon with motion-based spatial transformation, and conduct extensive experiments to demonstrate the effectiveness of Flareon.

**Strengths:**

1. The authors are very familiar with the related backdoor attacks and make a comprehensive review.
2. The authors present extensive experiments.
3. The English writing is very smooth.

**Weaknesses:**

1. Paper organization (not very important). I suggest the authors shortening the length of the background part, or putting some of them into Appendix. The core method of Flareon starts at the bottom of page 5, which makes me anxious since it takes a long time to reach the key part.
2. The difference between Flareon and Wanet (very important). I notice that the core operation of Flareon is to apply a motion-based transformation with a flow-field function, which is very similar with Wanet. In my opinion, the biggest difference is that Wanet manipulates both images and labels while Flareon manipulates only images, which makes the novelty of this work limited. Of course, the way of calculating parameters of flow-field function differs but this is just a slight difference. However, I am not sure if I understand correctly. I suggest the authors elaborating the difference in detail.
3. Confused opinions. The authors claimed that “None of the existing backdoor attacks can be easily adapted as code-injection attack without compromising the train-time stealthiness specifications”. However, in my opinion, we could adapt BadNets (or some other attacks) to code-injection attack as long as the trigger transformation function is put into the data augmentation module in some way like Figure 2(b).
4. Some notations. What does the notation “[1/H1/W]” mean? [1/H, 1/W] or 1/H\cdot 1/W ?Also, the dimension of \tau_y and [1/H1/W]^T does not match. I guess the authors mean \tau_y \odot torch.expand([1/H, 1/W]). Please clarify it.

I would reconsider my rating if the authors could address some of my concerns.

**Questions:**

Please refer to the weaknesses.